# Ideologies, Conspiracy Beliefs, and the Chinese Public's Politicized Attitudes to Climate Change

Yeheng Pan [1], Yu Xie [1,2], Hepeng Jia [1,2,*] and Xi Luo [1]

1. School of Communication, Soochow University, Suzhou 215127, China
2. Center for Science Communication & Scientific Innovation, Yangtze Delta Region Institute of Tsinghua University, Jiashan 314100, China
* Correspondence: hpjia@suda.edu.cn

**Abstract:** While ideologies consistently influence public opinions on climate change in Western democracies, whether they affect the Chinese public's climate attitudes is unknown. By applying a well-established measure of Chinese ideology, this study conducted a nationwide survey ($n$ = 1469) on the relationships between climate attitudes and ideologies, conspiracy beliefs, and science literacy. It is the first study to empirically investigate the impact of ideological tendencies, conspiracy beliefs, and conspiratorial thinking in shaping people's climate attitudes. Among a series of novel findings, ideology was found to be a crucial factor in Chinese attitudes toward climate change, and economic ideology, in particular, was most strongly related to climate attitude. Moreover, somewhat counterintuitively, we found a positive link between respondents' conspiratorial thinking and their climate awareness, as well as the failure of the moderation role of science literacy on ideological factors that influence climate attitude. All these findings suggest a mechanism behind the Chinese public's perception of climate change, primarily working on the individual–state relationship.

**Keywords:** climate change; ideology; conspiracy beliefs; science literacy; conspiratorial thinking





## 1. Introduction

Climate change is one of human beings' most prominent challenges in the new century. As the world's largest emitter of greenhouse gases, China plays a central role in fighting climate change [1]. The government recently pledged that it would reach carbon peak by 2030 and be carbon neutral by 2060. The climate issue has accordingly become a salient issue in the policy agenda. In the process, it is essential to mobilize the public to support low-carbon policies and implement environmentally friendly actions. However, many factors influence public attitudes to climate change and their corresponding behaviors [2,3]. Among them, ideologies—a belief system that consists of a "configuration of ideas and perspectives in which the elements are bound together by some form of constraint or functional interdependence" [4]—have been found to consistently influence people's attitude to climate change, at least in the Western democracies [5–7].

Though a batch of studies have explored the Chinese public's attitudes to climate change [8–10], and occasionally they touch individualist-collectivistic distinctions [11–13], there hasn't been any study examining how ideological consideration may have impacted such attitudes in China. On the other hand, like other parts of the world, China is experiencing an increasingly polarized public opinion environment [14]. In China, ideological polarization has become increasingly apparent and has been found to influence judgments on technologies [15,16]. It is, therefore, meaningful to examine how ideological factors may have been linked to people's climate change attitudes.

Ideology can be understood as a filter to shape identities, worldviews, and belief systems related to climate change [15]. In Western countries, studies have observed that people on the left side consistently report stronger beliefs in climate change and support for

action to mitigate it than citizens on the right [17,18]. Politically, the ideological distinction involves contrasting views on the government. Conservatives reject government or state intervention to address climate change [19].

The logic for political partisanship also goes to the economic domain, with a free market worldview tending to deny that climate mitigation is necessary [20,21]. The underlying logic is the free-market worldview opposes any government intervention, including those for improving the environment. Cultural values matter too. Conservatives favor human dominance over (and the exploitative utilization of) nature more than liberals [22]. Therefore, conservatives oppose environmental protection causes, including those addressing climate change.

Can the ideological imprint on people's climate attitude be replicated in China, where ideological divides are increasingly apparent? In one sense, the ideological left–right division in China is primarily related to attitudes toward state power, with leftists favoring a stronger nation, public ownership, and greater social equality [23,24]. This view seems to converge with the liberal thoughts in the West. But there isn't any literature exploring the left–right distinction of environmental protection in China. It is the task of the current study to fill the gap.

There are clues for tracing the role of cultural values in influencing people's climate attitudes. Traditional Chinese values do indeed respect nature [25], which means higher traditional value holders in China may be more likely to support addressing climate change if climate change is framed as a violation of nature. Recent studies have found that Chinese people with higher communitarian or egalitarian worldviews perceived more significant climate change risks and demonstrated more extensive support for climate policy [26]. These studies highlight the value of the current study.

In addition to ideologies, in the past decade, the rejection of any climate change consensus has been increasingly linked to conspiracy theories. Meanwhile, scholars have consistently found an association between ideologies and conspiracy beliefs. For example, the free-market worldview is more often connected with the rejection of climate change and vaccines, and people holding this view are more likely to consider them conspiracy theories coined by the elites [27,28]. Conservatives in the United States were more likely than liberals to endorse specific conspiracy theories, including those about climate change [29,30]. After the COVID-19 pandemic broke out, conservative ideology and media use were found to predict conspiracy beliefs [31]. Further studies concluded that pandemic denialism shared a similar pattern with climate denialism [32].

Studies also found conspiracy beliefs linked to the Chinese public's preventive measures against COVID-19 [33–36]. Meanwhile, Luo and Jia [37] found that Chinese respondents' conspiracy beliefs were closely related to their nationalism views—people with higher nationalism tended to believe the conspiracy beliefs that COVID-19 was born in the West. Similarly, there was also a conspiracy theory spread among the Chinese public that climate change was a Western plot to curb the development of China and other developing countries [38]. These pieces of evidence justify an investigation of the interlaced impacts of conspiracy beliefs and ideological views on the Chinese public's climate change attitudes.

Together with conspiracy beliefs, scholars often examined conspiratorial thinking (also called conspiracy ideation), which represented people's feeling of the world and its powerful actors as incomprehensible and the government's actions being full of secret plots [28,39,40]. Many studies found people who believe in one type of conspiracy theory tend to accept others, and the consequence of conspiratorial thinking is often consistent with conspiracy beliefs for predicting climate change attitudes [30,41].

Luo and Jia [37] found science literacy—a measure used to assess both scientific knowledge and analytical thinking—reduced COVID-related conspiracy beliefs among Chinese respondents, although only slightly. Elsewhere, it was found that reasoning ability and basic scientific knowledge were broadly associated with proscience views [42]. However, Drummond and Fischhoff [43] identified that people with greater science literacy and education have more polarized beliefs on controversial science topics. When examining

ideological and conspiracy impacts on Chinese people's climate attitude, what role may science literacy have played?

Based on the above theoretical reasoning, this study aims to investigate the relationship between the Chinese public's climate attitudes and ideological factors. Correspondingly, our general research questions are (1) whether economic, cultural, and political ideologies, conspiracy beliefs, and conspiratorial thinking can predict climate attitude, and (2) whether the relationships between climate attitudes and ideologies and conspiracy beliefs/thinking are moderated by science literacy.

Answering these questions allows us to better understand those factors underlying public engagement with climate change issues, which are inherently relevant to sustainability. By revealing the underexplored ideological underpinnings of the Chinese public's climate attitudes, this study can also help prevent political polarization from harming sustainability goals in China. It will also deepen our knowledge about the role of sociopolitical values in the public's understanding of climate change and paves ways to better public participation.

The following method section will introduce how we conducted the survey, how we measured each variable, and how we analyzed the data. Next, in the results section, we will present the result of the descriptive analyses followed by inferential analyses. We will then discuss the findings and their link to previous work before the work is concluded with contributions and limitations.

## 2. Materials and Methods

### 2.1. Study Design and Measures

This study is based on a nationwide online survey administered in China between 25 April and 17 May 2022 by the Shanghai-based survey firm Diaoyanba. The current sample's location, age cohorts, and gender distribution matched the population demographics in the China Statistical Yearbook 2019. The school administration at a large research university in East China, to which the researchers of this project are affiliated, approved the research plan to offset the lack of an institutional review board for the social sciences. In the questionnaire, we stressed anonymity and privacy protection and allowed participants to exit anytime they felt uncomfortable. We eventually gathered 1469 valid respondents.

The online questionnaire employed in this study aimed to assess the following six points about the Chinese public: (1) attitudes towards climate change; (2) demographic information; (3) ideological tendencies in the economy, culture, and politics; (4) conspiracy beliefs and conspiratorial thinking; (5) science literacy, and (6) influences of ideologies, conspiracy beliefs and science literacy in people's climate attitudes.

Based on previous studies [3,44], we measured the attitudes towards climate change by asking participants on a seven-point scale (1 = completely disagreement to 7 = completely agreement) the extent to which they agree with six climate-related statements, e.g., "there is sufficient scientific evidence to prove that climate change exists and humanity must take urgent action". The Cronbach's alpha of these six questions is 0.815.

In China, it is hard to ask respondents to report their ideological inclination directly. Instead, based on the classical literature and a well-established measure—the Chinese Political Compass—which was initially set up in 2007 by Peking University political scientists and international collaborators [16,45], we assessed respondents' agreements with a group of typical statements reflecting right or left tendencies in economic, cultural, and political ideologies. Guttman split-half coefficients of the two statements on economic liberalism is 0.631, on economic socialism is 0.678, on cultural conservatism is 0.625, on political liberalism is 0.622, and on political authoritarianism is 0.636. Cronbach's alpha of three statements on cultural anti-traditionalism is 0.521.

Regarding beliefs in conspiracy theories, we again asked respondents on a seven-point scale about their agreement on seven science-related conspiratorial statements. The measure consisted of conspiracy theories about climate change and COVID, GMOs, etc. They came from real-world conspiracy claims or previous studies [36,37,46]. The Cronbach's alpha

of seven statements is 0.731. Besides, we assessed agreements on conspiratorial thinking, evaluating people's claimed incomprehensibility of the government or elites' actions. The Cronbach's alpha of the five statements is 0.882.

We selected 10 questions from a pool to measure respondents' science literacy based on well-established instruments used worldwide, which were also adopted in our previous studies [36,37,47]. All questions consisted of three options: the statement is 'wrong', 'correct', or 'I don't know'. The Cronbach's alpha of the 10 statements is 0.654. In the procedure of descriptive statistics, one point was assigned only for the correct answer and zero for the "wrong" and "don't know" answers.

We reported the English translation of the survey questions in the tables below. All questions were translated literally unless the literal translation may be incomprehensible to English readers. In this case, we slightly adjusted the translation without distorting the original meanings.

### 2.2. Statistical Analysis

The statistical analysis was performed using SPSS Version 28.0 (IBM Corp., Armonk, NY, USA). We first ran the Descriptive Statistics function to show respondents' demographic characteristics and their judgments on each variable. Then, we analyzed the relationship between participants' attitudes to climate change and their ideological tendency, conspiracy belief, conspiratorial thinking, and science literacy with the Hierarchical Linear Regression function of SPSS. A $p$-value $< 0.05$ was considered statistically significant.

## 3. Results

### 3.1. Demographic Characteristics

A demographic breakdown of the studied group is presented in Table 1. Most of those surveyed were relatively young (18–24 (28.5%) and 25–34 (26.9%)), male (50.9%), had completed college degree education (40.5%) and junior college education (27.2%) and earned less than CNY 200,000 (86.8%) per year (USD 1 = CNY 7.11). The sample includes the majority of China's administrative provinces, with 71% of participants living in cities and 29% in rural areas. The distribution of the samples within each province is consistent with the characteristics of the country's population as a whole.

**Table 1.** Distribution of demographic characteristics of the sample ($n$ = 1469).

| Variable | % ($n$) |
|:---:|:---:|
| **Gender** | |
| Male | 50.9 (747) |
| Female | 49.1 (722) |
| **Age** | |
| <18 | 4.7 (69) |
| 18–24 | 28.5 (418) |
| 25–34 | 26.9 (395) |
| 35–44 | 25.7 (378) |
| 45–54 | 13.1 (193) |
| 55–64 | 0.6 (9) |
| >64 | 0.5 (7) |
| **Education level** | |
| Junior high school and below | 9.9 (146) |
| Senior high school | 19.1 (280) |
| Junior college education | 27.2(399) |
| College degree | 40.5 (595) |
| Postgraduate degree | 2.7 (40) |
| Doctoral Degree | 0.6 (9) |

**Table 1.** *Cont.*

| Variable | % (*n*) |
|---|---|
| **Yearly income (CNY)** | |
| 100,000 or less | 59.2 (870) |
| 100,001–200,000 | 27.6 (405) |
| 200,001–500,000 | 9.6 (141) |
| 500,001–1,000,000 | 2.5 (36) |
| 1,000,001–5,000,000 | 0.8 (12) |
| More than 5,000,000 | 0.3 (5) |

### 3.2. Attitude to Climate Change

The survey data showed that the respondents were generally aware of climate change (see Table 2). An average of 44.0% agreed or totally agreed with the statements that climate change is scientifically founded, urgent, human-induced, relevant to all, and carries serious consequences. In particular, 47.5% agreed or totally agreed that "There is enough scientific evidence that climate change exists, and urgent action must be taken immediately", and 45.1% disagreed or totally disagreed that "Climate change is a natural development process of the earth, and the impact of human activities is small".

**Table 2.** Participants' attitudes to climate change (*n* = 1469).

| Question: Please Make Your Judgment on the Following Statements on Climate Change: | | | | | | | |
|---|---|---|---|---|---|---|---|
| | Totally Disagree % (*n*) | Disagree % (*n*) | A Little Disagree % (*n*) | Neutral % (*n*) | A Little Agree % (*n*) | Agree % (*n*) | Totally Agree % (*n*) |
| (1) There is enough scientific evidence that climate change exists, and urgent action must be taken immediately; | 2.2 (33) | 2.4 (35) | 3.5 (51) | 26.7 (392) | 17.7 (260) | 28.1 (413) | 19.4 (285) |
| (2) Some people's concerns about climate change are too exaggerated to warrant particular action (reversed coding); | 14.0 (206) | 21.3 (313) | 25.5 (375) | 29.4 (432) | 4.6 (67) | 3.2 (47) | 2.0 (29) |
| (3) Climate change is a natural development process of the earth, and the impact of human activities is small (reversed coding); | 18.5 (272) | 26.6 (391) | 21.0 (308) | 25.2 (370) | 4.2 (61) | 2.8 (41) | 1.8 (26) |
| (4) Climate change will lead to more infectious diseases; | 2.2 (32) | 2.8 (41) | 5.4 (80) | 29.5 (433) | 20.7 (304) | 26.9 (395) | 12.5 (184) |
| (5) If nothing is done, climate change will submerge many of our coastal cities; | 1.8 (26) | 1.5 (22) | 4.6 (67) | 29.3 (431) | 22.8 (335) | 24.5 (360) | 15.5 (228) |
| (6) Climate change response is irrelevant to ordinary people (reversed coding). | 34.3 (504) | 22.2 (326) | 11.7 (172) | 23.5 (345) | 4.6 (67) | 2.5 (37) | 1.2 (18) |
| **Total** Mean SD | | | | 5.12 1.00 | | | |

Notes: As statements 2, 3, and 6 are negative about climate change, we reverse their scores when calculating the mean of climate attitudes and the average percentage of positive attitudes.

### 3.3. Ideological Beliefs among Respondents

The ideological divide among the respondents was relatively blurred (see Table 3). The vast majority chose a neutral orientation. Even so, we can identify some tendencies. In terms of political ideology, respondents showed a more distinct trend toward political authoritarianism, although they did not oppose political liberalism. The most noticeable is that 35.1% agreed or totally agreed with the inapplicability of Western democracy to China (M = 4.89, SD = 1.326). One thing to note is that, in the West, liberals and conservatives are often equal to left-leaning and right-leaning politics, but in China, this is reversed. Liberals in the Chinese context are commonly more opposed to the government and are marked as right-leaning. In contrast, Chinese conservatives are often more left-leaning, given the nation's socialist ideology.

**Table 3.** Participants' ideological tendency towards politics, economy, and culture (*n* = 1469).

| Question: Please Make Your Judgment on the Following Statements: | | | | | | | |
|---|---|---|---|---|---|---|---|
| | Totally Disagree % (*n*) | Disagree % (*n*) | A Little Disagree % (*n*) | Neutral % (*n*) | A Little Agree % (*n*) | Agree % (*n*) | Totally Agree % (*n*) |
| **Political Liberalism** | | | | | | | |
| Human rights are above sovereignty. | 6.6 (99) | 7.0 (105) | 7.5 (113) | 47.7 (715) | 10.7 (161) | 11.2 (168) | 9.3 (139) |
| In the event of a major social security incident, the government should still open the dissemination of information, even if the disclosure of information will bring the risk of riots. | 5.5 (82) | 7.9 (119) | 12.7 (191) | 42.1 (632) | 14.2 (213) | 10.5 (157) | 7.1 (10.6) |
| **Political Authoritarianism** | | | | | | | |
| The mistakes made by Mao were insignificant relative to his merits. | 3.7 (55) | 4.7 (70) | 8.9 (133) | 48.5 (727) | 11.9 (178) | 13.4 (201) | 9.1 (136) |
| Western democracy does not apply to China. | 2.1 (31) | 0.9 (14) | 3.4 (51) | 42.9 (644) | 15.6 (234) | 20.4 (306) | 14.7 (220) |
| **Cultural Conservatism** | | | | | | | |
| Zhou Yi Bagua can effectively explain many things. | 7.4 (111) | 9.3 (140) | 12.1 (182) | 47.9 (718) | 12.3 (184) | 7.5 (112) | 3.5 (53) |
| Chinese medicine has a better concept of human health than Western (modern) medicine. | 2.5 (37) | 4.1 (62) | 6.3 (95) | 48.7 (730) | 16.7 (251) | 13.7 (205) | 8.0 (120) |
| **Cultural Antitraditionalism** | | | | | | | |
| As long as they wish, it is their freedom to have sex between two adults. | 6.0 (90) | 7.0 (105) | 10.0 (150) | 41.3 (619) | 15.4 (231) | 12.9 (193) | 7.5 (112) |
| If it is voluntary, I approve of my child being in a same-sex partnership. | 9.6 (144) | 9.9 (149) | 10.9 (163) | 42 (630) | 9.1 (136) | 11.1 (166) | 7.5 (112) |
| Sexism and stereotypes of women are prevalent in society, and women's affirmative action (me too) is necessary. | 4.0 (60) | 3.1 (47) | 4.5 (67) | 39.4 (591) | 16.0 (240) | 19.7 (296) | 13.3 (199) |
| **Economic Liberalism** | | | | | | | |
| Private individuals should be able to own and buy and sell land. | 15.9 (239) | 14.7 (220) | 11.3 (169) | 38.1 (571) | 8.5 (128) | 7.2 (108) | 4.3 (65) |
| The gap between rich and poor is reasonable, and there is no need to interfere; With the development of the economy, the gap between the rich and the poor will naturally become smaller and smaller. | 16.5 (248) | 20.9 (313) | 15.9 (238) | 31.5 (472) | 6.9 (103) | 5.7 (85) | 2.7 (41) |
| **Economic Socialism** | | | | | | | |
| If prices or housing prices are too high, the government should intervene and control them. | 2.1 (32) | 1.5 (22) | 4.3 (65) | 26.7 (401) | 15.2 (228) | 27.2 (408) | 22.9 (344) |
| All areas related to national security, critical national economy, and people's livelihood must be controlled by state-owned enterprises. | 3.3 (50) | 4.3 (64) | 9.7 (145) | 39.6 (594) | 13.1 (196) | 15.2 (228) | 14.9 (223) |

Culturally, the respondents showed the coexistence of conservatism and antitraditionalism. For example, the respondents affirmed the advancement of Chinese medicine over Western (modern) medicine (M = 4.46, SD = 1.294) while supporting gender equality simultaneously (M = 4.73, SD = 1.466) (modern medicine is widely called Western medicine in China). Regarding economic ideology, respondents tended to lean toward economic socialism. For example, 37.4% disagreed or totally disagreed with the legitimacy of the rich/poor gap, and 50.1% agreed or totally agreed with the government's intervention in high housing prices.

### 3.4. Belief in Science-Related Conspiracy Theories and Conspiratorial Thinking

The results (Table 4) showed that the respondents did not trust conspiracy theories highly (M = 3.73, SD = 1.16). Most respondents remained neutral on the conspiracy theory, except for dismissing the argument that COVID-19 came from a Wuhan laboratory in China. In particular, for two conspiracy theories about climate change (Statements 6 and 7), more respondents chose to disagree.

**Table 4.** Participants' belief in conspiracy theories and conspiratorial thinking (*n* = 1469).

| Question: Please Make Your Judgment on the Following Statements: | | | | | | | |
|---|---|---|---|---|---|---|---|
| | **Totally Disagree** % (*n*) | **Disagree** % (*n*) | **A Little Disagree** % (*n*) | **Neutral** % (*n*) | **A Little Agree** % (*n*) | **Agree** % (*n*) | **Totally Agree** % (*n*) |
| **Conspiracy Theories** | | | | | | | |
| The coronavirus was synthesized artificially by American companies. | 6.9 (103) | 8.6 (129) | 8.9 (133) | 37.1 (557) | 20.7 (311) | 9.0 (135) | 8.8 (132) |
| GMOs are a foreign conspiracy to control Chinese grain. | 9.3 (140) | 14.4 (216) | 17.1 (256) | 33.7 (505) | 13.6 (204) | 7.0 (105) | 4.9 (74) |
| COVID-19 came out of a laboratory in Wuhan. | 56.6 (849) | 17.9 (269) | 10.0 (150) | 14.9 (223) | 0.3 (4) | 0.3 (4) | 0.1 (1) |
| The landing of the American Apollo 13 spacecraft on the moon was faked. | 11.3 (169) | 14.1 (212) | 12.9 (38.3) | 46.5 (698) | 7.3 (110) | 4.1 (62) | 3.7 (55) |
| HIV originally came from a military laboratory in a Western country. | 8.3 (125) | 9.5 (143) | 11.5 (173) | 40.7 (611) | 15.0 (225) | 8.3 (125) | 6.5 (98) |
| The issue of global warming is a plot of developed countries to contain the rise of developing countries. | 17.5 (263) | 15.9 (239) | 16.9 (253) | 31.8 (477) | 9.2 (138) | 4.4 (66) | 4.3 (66) |
| Climate change can turn cold & dry land into warm and arable land. | 9.8 (147) | 16.9 (254) | 17.3 (259) | 34.9 (524) | 10.1 (151) | 7.3 (110) | 3.7 (55) |
| **Participants' Conspiratorial Thinking** | | | | | | | |
| The inside story of many essential things in the world is kept from the public. | 3.4 (51) | 3.1 (47) | 7.2 (108) | 36.6 (549) | 21.8 (327) | 16.9 (253) | 11.0 (165) |
| Officials often do not tell us the real motives behind their decisions. | 3.1 (47) | 3.8 (57) | 10.7 (160) | 37.7 (565) | 20.4 (306) | 15.3 (229) | 9.1 (136) |
| Government agencies closely monitor all citizens. | 7.3 (110) | 9.8 (147) | 20.1 (302) | 40.1 (602) | 11.1 (166) | 7.4 (111) | 4.1 (62) |
| Incidents that appear to be unconnected are often the result of covert activities. | 3.7 (56) | 3.9 (59) | 10.3 (154) | 42.3 (635) | 21.1 (317) | 11.5 (172) | 7.1 (107) |
| Some secret organizations have a significant influence on political decision-making. | 4.1 (61) | 3.7 (56) | 7.5 (112) | 41.1 (616) | 20.4 (306) | 14.2 (213) | 9.1 (136) |

We also set five questions to measure conspiratorial thinking, which previously published research has widely adopted [39,48]. Statistics also showed that people scored relatively high on conspiratorial thinking (M = 4.35, SD = 1.14).

### 3.5. Scientific Literacy

Respondents had relatively high science literacy. As reported in Table 5, the correct answers (the bold options) of all the scientific questions received the most votes. Figure 1 further shows the percentage of the scientific literacy scores, where a score of 10 means that the respondent answered all 10 questions correctly. We can see that those with a score of 6 or above in scientific literacy level accounted for 82.9% (*n* = 1469).

### 3.6. The Association between Ideology, Conspiracy, Scientific Literacy, and Attitudes

We used the hierarchical regression model to examine the relationship between public attitudes to climate change and a group of independent variables: demographic factors (gender, age, education, and income), knowledge factors (science literacy), conspiracy beliefs and conspiratorial thinking, and ideological tendencies in economy, culture, and politics. Hierarchical regression enabled us to observe whether the investigated factors explain the statistically significant variance in the dependent variable (DV) after accounting for all other relevant variables [49].

The regression result is presented in Table 6. For the demographic variables in model 1, there was a correlation between gender (Male = 0, $\beta$ = 0.09, $p < 0.001$) and attitudes toward climate change, and education ($\beta$ = 0.16, $p < 0.001$) was positively associated with participants' attitudes. The demographics block accounted for 3.2% of the variances.

In model 2, we found that political authoritarianism ($\beta$ = 0.11, $p < 0.001$), antitraditional culture ($\beta$ = 0.09, $p < 0.01$), and economic socialism ($\beta$ = 0.37, $p < 0.001$) were all positively associated with participants' attitudes toward climate change. However, there was a

negative association between economic liberalism and attitudes ($\beta = -0.21$, $p < 0.001$). This block added 25.3% to the variances in our dependent variable.

**Table 5.** Participants' scientific literacy ($n$ = 1469).

| General Scientific Questions | | | |
|---|---|---|---|
| **Question:** The doctor told a couple that because they have the same morbid genes, if they give birth to a child, their chance of genetic disease is 1/4. Do you think the following statement is correct? | Wrong % ($n$) | Correct % ($n$) | I don't know % ($n$) |
| If they have three children, none of them will get genetic diseases. | **88.3 (1297)** | 2.3 (34) | 9.4 (138) |
| If their first child has a genetic disease, the subsequent three children will not have a genetic disease. | **85.4 (1254)** | 3.3 (49) | 11.3 (166) |
| If the first three children are healthy, the fourth child must have a genetic disease. | **87.6 (1287)** | 1.7 (25) | 10.7 (157) |
| Their children may have genetic diseases. | 15.0 (220) | **75.9 (1115)** | 9.1 (134) |
| **Question:** Do you think the following statement is correct? | | | |
| Cov-SARS-2 can cause SARS and pneumonia, but it will not cause colds. | **79.4 (1167)** | 11.2 (164) | 9.4 (138) |
| Electrons are smaller than atoms. | 32.6 (479) | **46.8 (688)** | 20.6 (302) |
| The mother's genes determine whether the child is a boy or a girl. | **86.1 (1265)** | 8.0 (117) | 5.9 (87) |
| Lasers are produced by converging sound waves. | **38.7 (569)** | 28.2 (414) | 33.1 (486) |
| Antibiotics (such as penicillin, streptomycin, or cephalosporin) can kill bacteria and viruses. | **57.1 (839)** | 30.1 (442) | 12.8 (188) |
| If you eat genetically modified fruit, human genes may change. | **68.9 (1012)** | 16.7 (246) | 14.4 (211) |

Notes: the bold options are the correct responses to the questions.

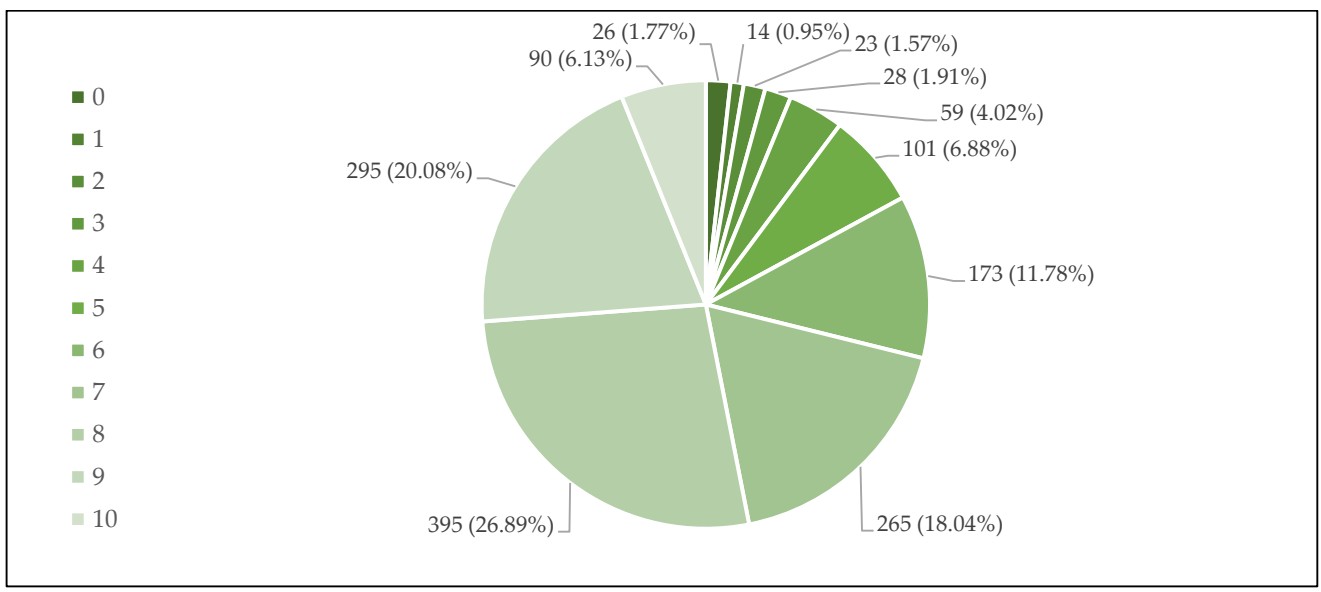

**Figure 1.** Level of the science literacy of the participants ($n$ = 1469).

In model 3, we found a negative association between conspiracy belief and attitudes to climate change ($\beta = -0.15$, $p < 0.001$) and a positive association between conspiratorial thinking and attitudes ($\beta = 0.22$, $p < 0.001$). This block added 4.5% to the variances in attitudes toward climate change.

In model 4, we tested the association between science literacy and climate attitudes. we found a positive association between them ($\beta = 0.16$, $p < 0.001$). The higher the participants scored in science literacy, the higher they scored in attitudes to climate change. The addition of science literacy enhanced the model's explanatory power by 2.0% for variance.

In order to present the moderation role of science literacy in the association between the different factors and attitudes toward climate change, we put eight interaction terms (by six types of ideology tendencies, conspiracy beliefs, conspiratorial thinking, and scientific literacy) into model 5. We found a negative moderating role for scientific literacy in the association between conspiracy beliefs and attitudes to climate change ($\beta = -0.05$, $p < 0.05$). Figure 2 shows that, among those with higher science literacy, the literacy's function to counterattack the negative influence of conspiracy beliefs in climate attitude declined more quickly than those with lower science literacy.

**Table 6.** Hierarchical regression model of the factors associated with attitude to climate change ($n = 1469$).

| Step | Variable | Public Attitude to Climate Change | | | | | | | | | |
|---|---|---|---|---|---|---|---|---|---|---|---|
| | | Model 1 | | Model 2 | | Model 3 | | Model 4 | | Model 5 | |
| | | $\beta$ | | $\beta$ | | $\beta$ | | $\beta$ | | $\beta$ | |
| 1 | Demography | | | | | | | | | | |
| | Gender (Male = 0) | 0.09 | *** | 0.07 | ** | 0.08 | *** | 0.07 | ** | 0.07 | ** |
| | Age | −0.01 | | −0.01 | | 0.00 | | 0.00 | | −0.01 | |
| | Education | 0.16 | *** | 0.07 | * | 0.06 | * | 0.03 | | 0.03 | |
| | Income | −0.05 | | −0.03 | | −0.02 | | −0.02 | | −0.01 | |
| 2 | Ideology | | | | | | | | | | |
| | Economic Liberalism | | | −0.21 | *** | −0.20 | *** | −0.17 | *** | −0.18 | *** |
| | Economic Socialism | | | 0.37 | *** | 0.35 | *** | 0.33 | *** | 0.33 | *** |
| | Cultural conservatism | | | −0.02 | | −0.03 | | −0.02 | | −0.01 | |
| | Antitraditional culture | | | 0.09 | ** | 0.05 | | 0.04 | | 0.03 | |
| | Political liberalism | | | 0.04 | | 0.02 | | 0.02 | | 0.02 | |
| | Political authoritarianism | | | 0.11 | *** | 0.11 | *** | 0.11 | *** | 0.11 | *** |
| 3 | Conspiracy | | | | | | | | | | |
| | Conspiracy Belief | | | | | −0.15 | *** | −0.13 | *** | −0.13 | *** |
| | Conspiratorial Thinking | | | | | 0.22 | *** | 0.21 | *** | 0.20 | *** |
| 4 | Scientific literacy | | | | | | | | | | |
| | Scientific literacy | | | | | | | 0.15 | *** | 0.16 | *** |
| 5 | Interactions | | | | | | | | | | |
| | Scientific literacy × Economic liberalism | | | | | | | | | 0.02 | |
| | Scientific literacy × Economic socialism | | | | | | | | | −0.01 | |
| | Scientific literacy × Cultural conservatism | | | | | | | | | −0.01 | |
| | Scientific literacy × Cultural Anti-tradition | | | | | | | | | 0.01 | |
| | Scientific literacy × Political liberalism | | | | | | | | | −0.01 | |
| | Scientific literacy × Political authoritarianism | | | | | | | | | −0.04 | |
| | Scientific literacy × Conspiracy belief | | | | | | | | | −0.05 | * |
| | Scientific literacy × Conspiratorial thinking | | | | | | | | | −0.01 | |
| | $n$ | 1469 | | 1469 | | 1469 | | 1469 | | 1469 | |
| | incremental $R^2$ | 3.5% | | 25.3% | | 4.5% | | 2.0% | | 0.6% | |
| | $R^2$ | 3.5% | | 28.8% | | 33.4% | | 35.4% | | 36.0% | |
| | adjusted $R^2$ | 3.2% | | 28.4% | | 32.8% | | 34.8% | | 35.1% | |

\* $p < 0.05$. \*\* $p < 0.01$. \*\*\* $p < 0.001$.

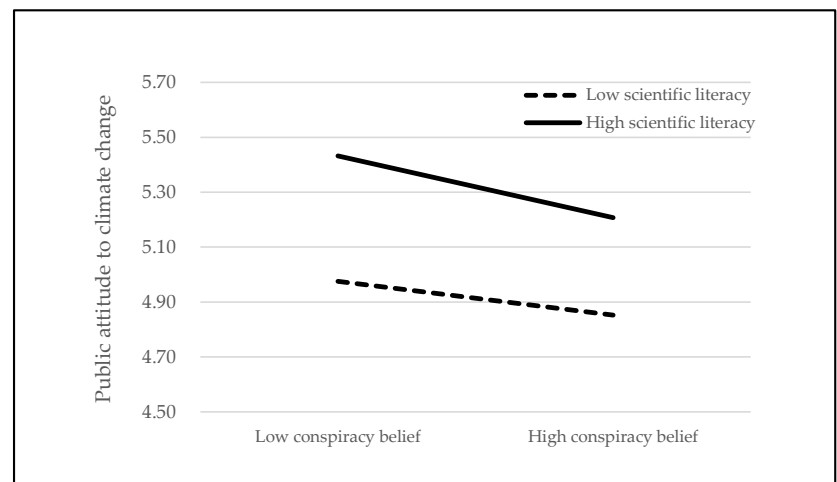

**Figure 2.** The moderating effect of scientific literacy on the association between conspiracy belief and attitudes.

## 4. Discussion

Our respondents generally had a high level of recognition of climate change and were usually scientifically literate. They had ideological tendencies within political, economic, and cultural aspects. Their beliefs in conspiracy theories were not so firm, but their conspiratorial thinking, which believes that government's and elites' behaviors are incomprehensible and secret, was relatively high. The recent decade has witnessed the Chinese government become increasingly proactive in dealing with climate change, either on the global stage or at home [50]. Chinese official media, e.g., the flagship party organ called *People's Daily*, has portrayed China as playing a responsible and contributing role in global climate governance since Copenhagen 2009 [51]. As the officials set the tone, those skeptical voices that used to be loud on social media lowered their volume [52]. All of this might have led to the Chinese public's high recognition of climate change, which also seemed in line with the two national climate surveys conducted by the quasi-official China Center for Climate Communication [53].

Regarding our main research target: ideology, we found ideology crucial for influencing people's attitudes to climate change. Notably, economic ideology was most strongly related to climate attitudes, with left-leaning values more accepting of climate change facts and their impacts. At the same time, the right-leaning views tended to disagree with climate change statements. This finding is consistent with previous studies conclusions on the consequences of the free market worldview [20,21].

Cultural and political ideologies had a much weaker connection with climate attitudes. Antitradition cultural values were initially weakly linked to climate attitudes, but the significance disappeared after we added conspiracy beliefs. Left-leaning political ideology was significantly associated with positive climate attitudes in a constant mode. However, we must be cautious in concluding it replicates the left–right distinction in climate change found in the US and Europe [6,21]. It is possible that left-leaning political ideology may endorse the Chinese government's decisions more than those of political liberalists [14,54], and thus they adopt a positive attitude to addressing climate change.

To our surprise, respondents' beliefs in various conspiracy theories and their conspiratorial thinking were relatively strongly linked to their attitudes to climate change but in different directions. Conspiracy beliefs were negatively associated with attitude to climate change, which is in line with most studies in the West [27,30]; however, conspiratorial thinking positively predicted people's climate attitude. The pattern was relatively stable across different models, and the addition of science literacy did not change the pattern and the predictive power of the variables. The negative association between conspiracy beliefs may be explained by the fact that many conspiracy theories listed by us—primarily selected from rampant ones on China's social media—were anti-West or at least targeting foreign institutions. In contrast, a primary version of climate conspiracy theory in China is that global warming is a plot of developed countries to contain the rise of developing countries [38]. On the other hand, the positive prediction of conspiratorial thinking might lay in its acknowledgment of the incomprehensibility of the government's actions. This conviction could converge with the Chinese public's embrace of authoritarianism or statism [55].

As expected, science literacy relatively strongly predicted positive attitudes to climate change. But its moderation role was fragile. It can only weakly moderate the reversed direction of the consequences of conspiracy beliefs. The higher people's science literacy was, the less likely their conspiracy beliefs would weaken their climate change attitudes. While this is coherent with one previous study [42], science literacy failed to moderate the impact of ideological beliefs and conspiratorial thinking. This might be caused by the stronger political imprint in the Chinese mind, which shaped their views more than scientific knowledge or the reasoning covered by science literacy. Recent studies also found that discussions about COVID-19 science did not take a prominent position on Weibo, suggesting a lack of effective science and risk communication [56]. We observed similar patterns for climate change communication on Chinese social media.

Even the weak role of science literacy in moderating the consequence of conspiracy beliefs on climate attitudes was further attenuated, as shown in Figure 2. The figure shows that, among those with higher science literacy, the function of literacy to counterattack the negative influence of conspiracy beliefs for climate attitude declined more quickly than those with lower science literacy. When taken together, these results reminded us to rethink the role of science literacy, conspiracy beliefs, and conspiratorial thinking in shaping and adjusting our attitude to climate change.

## 5. Conclusions

This novel study revealed the consequences of ideological beliefs in climate change attitudes in China for the first time. It is also the first study to empirically investigate the impact of conspiracy beliefs and conspiratorial thinking in shaping people's climate attitudes in the country.

The current study found that economic ideological beliefs strongly predicted the Chinese public's attitude to climate change. Economic liberalism view holders disagreed with climate change claims, while economic socialism supported actively dealing with climate change. Conspiracy beliefs negatively predicted people's climate attitudes, but conspiratorial thinking was positive in this regard. Although science literacy was positively associated with climate attitudes, its moderation role in offsetting the impact of "undesirable determinants" was minimal.

These findings are novel and significant in demonstrating the politicized features of the Chinese public's climate perception. However, being the first such study, the results and implications need further similar studies to verify them.

Despite the lack of similar studies, the current research has significant policy implications. First, knowing the vital role of ideological influence in shaping people's climate attitudes can help climate change campaigns more effectively mobilize the public by stressing the sociopolitical consequences of dealing with global warming. Second, as mentioned above, ideological polarization has jeopardized people's cognition, attitude, and behaviors regarding climate change in the West. Revealing the role of ideological beliefs in China can prevent the undesired behavioral consequences resulting from such polarization. Third, studies like global warming's Six Americas provided a pathway for selective behavioral interventions [57]. In these studies, ideology is one of the primary factors underlying different population sections. Our study may also offer insights into initiating similar intervention strategies in China. Finally, sustainability policymaking requires dialogue and debate among different research areas [58]. Our study can more constructively fuel the discussion by bringing in a new dimension, influencing the Chinese people's climate attitude and behaviors.

Finally, our study is not without limitations and invites extensions. Firstly, this study is mainly based on a cross-sectional survey, so we must be cautious in reaching any causational conclusion. Second, the sample size is small compared with China's huge population. Third, the standardization of some variables needs to be further improved. However, we do not think these limitations reduced the validity of this study. The politicization of Chinese people's attitudes and the intention for healthy behaviors are visible in many studies [34,37,59]. It is highly legitimate to extend the influences of politicization to environmental behaviors in China. Follow-up research might address this limitation with new data and methods while exploring the political aspects of Chinese people's environmental behaviors and attitudes.

**Author Contributions:** Conceptualization: Y.P., Y.X., H.J.; Methodology: Y.P., X.L., H.J.; data analysis: Y.X., X.L.; writing: Y.P., H.J., Y.X.; review and editing: Y.X., Y.P., H.J. All authors have read and agreed to the published version of the manuscript.

**Funding:** This research was funded by the Social Science Foundation of Jiangsu Province in China (No.22XWC001), Youth Project "Strategy and Effect of counter-framing in the international communication of climate change issues" of the National Social Science Foundation of China (No. 22CXW020), the Key Project "Study on the permanent mechanism of communicating scientific spirit and professionalism in the digital era" of the National Social Science Foundation of China (No.21AZD013), and General Project "Application of counter-framing in the international communication of low carbon" of the Education Department of Jiangsu Province in China (No. 2022SJYB1439).

**Institutional Review Board Statement:** The study was conducted according to the guidelines of the Declaration of Helsinki and approved by the School of Communication, Soochow University.

**Informed Consent Statement:** Informed consent was obtained from all subjects involved in the study.

**Data Availability Statement:** Materials and anonymous data are available from the authors by request.

**Conflicts of Interest:** The authors declare no conflict of interest.

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
