# Peer review of "Ideologies, Conspiracy Beliefs, and the Chinese Public’s Politicized Attitudes to Climate Change"

_sustainability, doi:10.3390/su15010131_

Round 1

Reviewer 1 Report

I apprecciate the author's approach in order to investigate the relationship between Ideologies, Conspiracy Beliefs, chinese politicized attitude and climate change. The sample is consistent more than 1400 interviews. 

But the paper present some major concern which could be reduce the qualiry of the article;

Introduction: Is to short is necessary to improve the climate change impact and the relationship between public policy and climate chenge; In addition in the final part of this section is necessary to specify the aim of the study and the paper structure.

In this sense see anc cite:

Fusco, G. (2021). Twenty Years of Common Agricultural Policy in Europe: A Bibliometric Analysis. Sustainability13(19), 10650.

in this paper there isn't a literature review section, i suggest to improve this consideration in the first part of the introduction.

the methodolofy section is correct and the choice of the questions are in line with the aim of this study and are "Valid" like demonstrate the crinbach's alph test.

But in my opinion is necessary to explain better the different step and question in the methodology section, please improve this aspects.

Result section: This section is clear and well written, the discussion are in line with the results and this aspects represent the strong of this paper.

Conclusion: Improve the aspects of the policy implication

Author Response

Our point-to-point response to Reviewer1:

  1. “Introduction: Is to short is necessary to improve the climate change impact and the relationship between public policy and climate chenge; In addition in the final part of this section is necessary to specify the aim of the study and the paper structure. In this sense see anc cite: Fusco, G. (2021). Twenty Years of Common Agricultural Policy in Europe: A Bibliometric Analysis. Sustainability13(19), 10650.”

Response: Thank you for the very instructive suggestions. As suggested by the reviewer, we added sentences about the research goal and structure of the paper at the end of the introduction section (please see line 119-136). As for the relationship between policy and climate change, we take this into consideration when modifying the entire paper, not merely in the introduction. We put China’s recent climate policy as the background of this study in the introduction so that readers can better understand the relevance of this study.

2.“in this paper there isn't a literature review section, i suggest to improve this consideration in the first part of the introduction.”

Response: Thank you for the suggestion. We understand the concern of the reviewer, but the general style of Sustainability is to use Introduction to provide justification of the research reasoning and current research questions. In other words, the current introduction, in our opinion, has served the purpose of a literature review which is to bridge the research background and more specific research questions based on the literature. make dramatic changes and although the other two reviewers seem fine with the current format, we still carefully went through the texts and made adjustments wherever necessary (for example, we added a paragraph on research questions with relevant wording). We hope this defense could be acceptable, but in the meantime, we are open to further modification.

  1. “the methodolofy section is correct and the choice of the questions are in line with the aim of this study and are "Valid" like demonstrate the crinbach's alph test. But in my opinion is necessary to explain better the different step and question in the methodology section, please improve this aspects.”

Response: Thanks for the instruction. We revised the Statistical Analysis section and explained the two steps of analysis.

  1. “Conclusion: Improve the aspects of the policy implication”

Response: Thanks for the suggestion. Now we’ve emphasized the policy implication in the conclusion. 

Reviewer 2 Report

Dear Authors,

Thank you for your paper. The paper is well structured and presented in an easy understandable way. The list of references is substantial. The topic addressed is of wide interest. It is appreciated that you took into account simultaneously conspiracy beliefs and attitudes to climate change. It is also appreciated that you included reverse coding in your set of questions in order to cross check the answers.

Of course, tradition and ideology can have significant impact. While in the West individualism is predominant, in China it might still be more present collectivism. From this point of view the question is what is the government doing to influence people’s attitude towards climate change? Is there a strong official position, is climate change one of the main themes in the mass media? Is there a discrepancy between reporting on climate change in official media and on social platforms? How often is the population informed on severe weather/hydro events and potential increase in frequency of such events due to climate change?

What is missing is more information on how the respondents were selected. Were respondents mostly living in highly industrialised areas where environment pollution is an issue or in areas with higher exposure to severe weather events? It might be also of interest to know in which kind of institutions they are employed? Were they invited to fill the questionnaire and how many refused taking part in this research? Were they living in big cities, were participants from more rural parts also included?

As you already noticed, the sample is small. As somebody who knows China only as an occasional visitor, information about government campaigns on climate change mitigation needs would help to better understand the results. Are in China civil initiatives or groups advocating for climate change mitigation?

Is climate change anyhow addressed during formal education?

It might be that there is a significant discrepancy between belief and attitude in climate change science and individuals’ preparedness to act or change their behaviour in order to mitigate climate change. We notice this a lot in the West. Therefore, it is recommended to investigate this aspect.

In line 73 there must be something missing in the sentence: “In China, Liu (31).”

Best wishes

Author Response

Our point-to-point response to Reviewer2

  1. Reviewer 2: “Of course, tradition and ideology can have significant impact. While in the West individualism is predominant, in China it might still be more present collectivism. From this point of view the question is what is the government doing to influence people’s attitude towards climate change? Is there a strong official position, is climate change one of the main themes in the mass media? Is there a discrepancy between reporting on climate change in official media and on social platforms? How often is the population informed on severe weather/hydro events and potential increase in frequency of such events due to climate change?”

Response: we agree that the reviewer has pointed out a list of important aspects regarding the Chinese public perception of climate change. To improve this paper, we’ve now incorporated these important insights into our discussion section. We also understand that a thorough answer to these questions should be in several individual papers. Actually, our co-authors have published on issues like the Chinese media framing of climate change and we have a paper under review on the influence of institutional trust on public climate attitudes. We also would like to better answer them in future research.

  1. “What is missing is more information on how the respondents were selected. Were respondents mostly living in highly industrialised areas where environment pollution is an issue or in areas with higher exposure to severe weather events? It might be also of interest to know in which kind of institutions they are employed? Were they invited to fill the questionnaire and how many refused taking part in this research? Were they living in big cities, were participants from more rural parts also included?”

Response: Thanks for the suggestion. As we explained in the text, the current sample’s location, age cohorts, and gender distribution matched the population demographics in the China Statistical Yearbook 2019. We didn’t select the respondents by ourselves but commissioned the survey firm Diaoyanba to select respondents from its pool in accordance with the statistical yearbook. This procedure, as well as the company’s data, have been accepted by previous publications and thus we believe in its validity. We surveyed more about demographic information than what we showed, so now we added more information of interest in the results section.

  1. “As you already noticed, the sample is small. As somebody who knows China only as an occasional visitor, information about government campaigns on climate change mitigation needs would help to better understand the results. Are in China civil initiatives or groups advocating for climate change mitigation? Is climate change anyhow addressed during formal education?”

Response: Yes, climate change and Chinese climate policy have been put into Chinese formal education as we know, and there are NGOs working in this field to be sure. The first author used to study climate NGOs in China and has published work on this topic. We also asked about the public trust in NGOs in this project; but for this study, as they are not the focus, we chose not to mention NGOs.

As for the relatively small sample against the large Chinese population, we would argue that the main purpose of this study is looking into the relationship between climate attitudes and socio-political psychological factors. For this purpose, the size of the current sample should be okay for us to make reliable statistical inferences about the relationship.

  1. “It might be that there is a significant discrepancy between belief and attitude in climate change science and individuals’ preparedness to act or change their behaviour in order to mitigate climate change. We notice this a lot in the West. Therefore, it is recommended to investigate this aspect.”

Response: Thank you! This is exactly on our research agenda for the future. This discussion is so important that we think it should be translated at least an individual paper.

5.“In line 73 there must be something missing in the sentence: “In China, Liu (31).””

Response: Thanks for pointing this out. This broken sentence has been deleted. We fixed a few typos apart from this.

Reviewer 3 Report

Introduction:

In the Introduction:

(1) State the main research question.

(2) Say why (a) this study important and (b) why this is of interest to the readers of this journal. (Please ensure what you say in (b) is relevant to persons with research interests in sustainability anywhere in the world.).

(3) Say how your study answers your research question.

(4) Add the contribution of your study.

Materials and methods

How did you choose to base this study in China? As the you state, “Due to China's authoritarian rules, it is hard to ask respondents to report their ideo-115 logical inclination directly.” Also, China is quite large, is it relevant to have a sample from the entire country?

Results

I think that there is a mistake in figure one. Maybe these are numbers, not percentages.

Discussion

The discussion part should be deepened. In the discussion section it is required to explicitly link back to the literature review. It is necessary to compare and contrast the findings with those that already exist in the literature. What is different? What is the same? How does the study add to better explanations of previously identified phenomena? How does it for example solve existing contradictions in findings of other studies? Or, how does it explain previously unexplainable findings? The number of references here is very small.

Conclusion

Restate the contribution of the paper. The theoretical contributions can be improved. There is no assessment as to whether this study is impactful or sustainable. Should the government do something, which is the bottom line?

Author Response

Our point-to-point response to Reviewer3:

  1. “In the Introduction: (1) State the main research question. (2) Say why (a) this study important and (b) why this is of interest to the readers of this journal. (Please ensure what you say in (b) is relevant to persons with research interests in sustainability anywhere in the world.). (3) Say how your study answers your research question. (4) Add the contribution of your study.”

Response: Thanks for the instruction. We’ve modified the introduction accordingly.

  1. “How did you choose to base this study in China? As the you state, “Due to China's authoritarian rules, it is hard to ask respondents to report their ideological inclination directly.” Also, China is quite large, is it relevant to have a sample from the entire country?”

Response: We study China for its being the largest carbon emitter in the world and its crucial role for addressing climate change. There have been widespread observations on how ideological separation/polarization affected climate attitude, but there isn’t any such study in China. Given the visible ideological separation/polarization in other fields in the country, it is necessary to examine the polarization’s relationship with the attitude. This is the basic reasoning of our research. In China, it is difficult to directly examine ideological inclination, so we adopted questions to calculate this. This has been done by many previous studies, mostly in the field of China study.

As for our sample, we tried to have a representative national sample in terms of the regional population, age, and gender, although it is never easy to reach a strictly representative sample. But as we responded to Reviewer 2, the purpose of our sampling is to test the association between ideological inclination and climate attitude rather than accurately and statistically reflect the climate attitudes and values of the Chinese population, and as our sampling procedure has been verified by previous publications, we believe our sample should be acceptable. In terms of statistics, the size of the current sample should be okay for us to make reliable statistical inferences about the relationship between climate attitudes and ideological factors. We hope this argument would be acceptable to the reviewer.

3.“I think that there is a mistake in figure one. Maybe these are numbers, not percentages.”

Response: Now, what is shown in Figure one is “number (percentage)”.

  1. “The discussion part should be deepened. In the discussion section it is required to explicitly link back to the literature review. It is necessary to compare and contrast the findings with those that already exist in the literature. What is different? What is the same? How does the study add to better explanations of previously identified phenomena? How does it for example solve existing contradictions in findings of other studies? Or, how does it explain previously unexplainable findings? The number of references here is very small.”

Response: Thanks for the suggestion! They indeed are essential parts of a decent discussion. Now we’ve revised the discussion part. We added relevant literature and also enhanced the discussion with links to previous studies. 

  1. “Restate the contribution of the paper. The theoretical contributions can be improved. There is no assessment as to whether this study is impactful or sustainable. Should the government do something, which is the bottom line?”

Response: We restated our contribution in the first two sentences of the conclusion and also, improved the discussion about the policy implications.

Round 2

Reviewer 1 Report

Well done! Now the paper is suitable for a pubblication!